# MR-YOLO: An Improved YOLOv5 Network for Detecting Magnetic Ring Surface Defects

**DOI:** 10.3390/s22249897

**Published:** 2022-12-15

**Authors:** Xianli Lang, Zhijie Ren, Dahang Wan, Yuzhong Zhang, Shuangbao Shu

**Affiliations:** 1Anhui Province Key Laboratory of Measuring Theory and Precision Instrument, School of Instrument Science and Opto-Electronics Engineering, Hefei University of Technology, Hefei 230002, China; 2Suzhou Institute of Biomedical Engineering and Technology, University of Science and Technology of China, Suzhou 215163, China

**Keywords:** object detection, magnetic ring, defect detection, MR-YOLO, YOLOv5, MobileNetV3

## Abstract

Magnetic rings are widely used in automotive, home appliances, and consumer electronics. Due to the materials used, processing techniques, and other factors, there will be top cracks, internal cracks, adhesion, and other defects on individual magnetic rings during the manufacturing process. To find such defects, the most sophisticated YOLOv5 target identification algorithm is frequently utilized. However, it has problems such as high computation, sluggish detection, and a large model size. This work suggests an enhanced lightweight YOLOv5 (MR-YOLO) approach for the identification of magnetic ring surface defects to address these issues. To decrease the floating-point operation (FLOP) in the feature channel fusion process and enhance the performance of feature expression, the YOLOv5 neck network was added to the Mobilenetv3 module. To improve the robustness of the algorithm, a Mosaic data enhancement technique was applied. Moreover, in order to increase the network’s interest in minor defects, the SE attention module is inserted into the backbone network to replace the SPPF module with substantially more calculations. Finally, to further increase the new network’s accuracy and training speed, we substituted the original CIoU-Ioss for SIoU-Loss. According to the test, the FLOP and Params of the modified network model decreased by 59.4% and 47.9%, respectively; the reasoning speed increased by 16.6%, the model’s size decreased by 48.1%, and the mAP only lost by 0.3%. The effectiveness and superiority of this method are proved by an analysis and comparison of examples.

## 1. Introduction

A magnetic ring is a ring-shaped magnet that is widely used in common components in electronic circuits, and its quality directly affects the performance of the device. However, in the production process, due to the influence of raw material composition, processing technology, and equipment conditions, some defects, such as cracks and adhesion, will occur on the surface of the magnetic ring, which will affect the appearance and performance of the magnetic ring. Currently, most magnetic rings are tested by manual inspection, which has low detection efficiency and slow speed, and it is difficult to achieve intelligent production [1]. Therefore, there is an urgent need for new key technologies that can quickly and automatically detect magnetic rings in the industry.

In the past, manual visual inspections or non-destructive detection techniques were used to detect magnetic ring defects, which could avoid secondary pollution caused by contact detection. Common non-destructive testing techniques include ultrasonic testing technology [2], spectrum detection technology [3], laser detection technology [4], X-ray detection technology [5], etc., which are high in cost and have low efficiency and accuracy. It is challenging for a single technique to find all defects in a magnetic ring due to the diversity of defects. Meanwhile, machine vision techniques are employed to overcome these drawbacks since they make it simpler to implement automated production and information technology across a range of industries [6,7,8,9,10,11].

Machine vision uses sensors to capture images, extract features, and interpret them to obtain information for controlling machinery or processes. Nowadays, magnetic ring defect detection methods based on machine vision can be divided into two classes: traditional machine vision and online detection methods based on deep learning. The conventional detection method is that researchers design algorithms according to the types and characteristics of magnetic ring defects. For example, we can locate the defects by analyzing the grey difference between defect areas and regular areas in the magnetic ring surface image. Li et al. [1] proposed a magnetic ring defect detection method based on a mask image, which can accurately and quickly extract the defects in each area of the magnetic ring surface image. Aiming at the problem in which the conventional segmentation algorithm had difficulty extracting defects from complex textures, Hou et al. [12] adopted an edge segmentation algorithm based on the wavelet analysis of weakening texture and the adaptive Canny algorithm. To enhance the defect recognition ability, according to the circular image observation, Cui [13] used an adaptive segmentation algorithm based on a multi-scale, multiple-direction Gabor filter group, which can enhance the contrast of the defect. Wang [14] carried out a machine vision-based magnetic ring surface defect detection system, in which the defects were extracted according to the defect features. Then, the support vector machine (SVM) method was used for defect classifications.

The defect detection method based on deep learning has become the mainstream method in recent years, and it can automatically extract defect features and avoid complex algorithm designs. Deep learning methods can be classified into two types: two-stage methods and one-stage methods. The two-stage method involves object region proposal with deep networks, followed by object classification based on features extracted from the proposed region with bounding-box regression. The one-stage method predicts bounding boxes over the images without the region proposal step. It comprises an end-to-end target detector, including YOLO and SSD, which consumes less time and can be used in real-time applications. Li et al. [15] established a dataset of six types of surface defects on steel strips, improved the YOLO network, and applied it to the production line. Zhang et al. [16] modified the original YOLOv3 by introducing a novel transfer learning method with fully pretrained weights from a geometrically similar dataset and increasing the accuracy of concrete bridge surface damage detection. Chen et al. [17] tried to use DenseNet instead of YOLOv3’s Darknet-53 backbone network to detect SMD LED defects and achieved good results. Guo et al. [18] introduced a MSFT-YOLO model to detect defects of steel surfaces by adding the TRANS module. Wang et al. [19] proposed a YOLOv5 algorithm based on the improved MS-YOLOv5 model to detect the surface defects of aluminum profiles. By replacing the neck part of the original algorithm with a PE-Neck structure, the model’s ability to extract and locate defects at different scales was enhanced. Liao et al. [20] replaced the FPN structure of YOLOv5 with BiFPN structure for the surface defect detection of turbine blades, achieving higher-level feature fusion. However, With the deepening of the deep network, the computing overhead of the system increases; due to the popularity of edge computing and because of the neural network framework, YOLO is not capable of detecting minor defects, which can easily cause missed inspections. Therefore, this paper conducts an improved model based on YOLOv5.

In order to achieve good accuracy and speed, the YOLOv5s backbone network was replaced with MobileNetV3 for feature extraction. Some defects of the magnetic ring are relatively small, and it is easy to miss during inspections. To avoid missed detection, we introduce an effective SE module into the backbone network, which makes the model pay more attention to the primary information to improve detection accuracies. Moreover, we design an EIoU loss function to improve the localization accuracy of the model. The main contributions of this paper are as follows:The paper proposes a lightweight detection algorithm named MR-YOLO (YOLOv5 for Magnetic Ring) by replacing the backbone of the original YOLOv5 with the backbone of the lightweight MobileNetV3.We add the SE attention mechanism and introduce the updated SIOU-loss function into the model to improve the detection effect and expression effect of the model.The training dataset is enhanced with Mosaic data, and a GPU can generate more significant results, lowering the need for large mini-batch sizes.

All these works are tested and verified on the existing magnetic ring defect dataset, proving the proposed algorithm’s feasibility.

## 2. Related Work

### 2.1. Traditional Defect Detection Methods

Compared with the deep learning object detection method, the traditional surface defect detection method is mainly based on image processing. The general steps are as follows: image preprocessing, image segmentation, feature extraction, and classification. Medina et al. [21] proposed a rotation-invariant Gabor filter for defect image processing, which can detect defects in various directions. Ma et al. [22] designed a method for detecting the internal defects of integrated circuits based on wavelet transforms. Zhang et al. [23] developed a computer vision-based system for inspecting strongly reflected metal surface defects based on wavelet transforms, spectral measures, and support vector machines. The results demonstrated that the proposed method could identify seven classes of metal surface defects effectively. Meanwhile, the above examples demonstrate that traditional surface defect detection methods also have inherent drawbacks. Traditional surface defect detection methods mainly rely on artificially designed feature extractors, high-quality professionals, and complex parameter adjustment processes. For each of the above defect detection scenarios, corresponding specific algorithms must be developed with poor generalization ability and robustness. Nowadays, the target detection algorithm based on deep learning is mainly based on data-driven feature extractions. Deep sample features can be obtained according to the learning of a large number of samples, which has more advantages than the traditional surface detection algorithm. These gaps limit the application of traditional image processing methods.

### 2.2. Deep Learning Methods

Traditional image detection algorithm design is highly dependent on developers and researchers. There are also some shortcomings, such as long project cycles, poor robustness, low detection efficiency, and low detection accuracy. With the rapid development of deep learning object detection, more researchers have begun shifting the focus of magnetic ring defect detection to online detection technology based on deep learning. For example, Xiang et al. [24] built a pavement crack detection system by adding a transformer module to the YOLOv5 network and achieved high F1 scores. Li et al. [25] proposed an infrared image object detection network named YOLO-FIRI, which introduced an improved attention module in the residual block to focus on objects and to suppress the background, and it improved small object detection accuracies by adding multiscale detections. Similarly, Wang [26] used a new feature enhancement module (FEM) for the problem of missing features in fog scene images and utilized an attention mechanism to help the detection network pay more attention to the more useful features in the fog scenes. According to the characteristics of remote sensing images, Wang et al. [27] adopted a circular smooth label (CSL) method to calculate the loss of the rotating object detection bounding box and used the FcaNet attention mechanism to design new feature fusion modules. Chen et al. [28] combined Cycle-GAN and YOLOv5 to detect petrochemical pipelines, improving the efficiency and accuracy of detection.

Furthermore, due to the real-time requirements of magnetic ring detection, more researchers have been fond of improving the one-stage YOLO detection algorithm series [29,30,31,32]. For example, in 2020, Zhang [33] embedded the CBAM attention mechanism module in the YOLOV3 network and pruned its BN layer with a sparsification training strategy to reduce the model’s parameters and model volume to one-sixth of the original. In 2021, Wang [34] used an improved YOLOv3 algorithm to detect magnetic ring defects, in which a spatial pyramid pooling module was added to enrich the defect features. Similarly, Ying et al. [35] proposed modified-YOLOv5s to detect defects of wire-braided hoses, and the effect of efficient channel attention mechanism (ECA) in different positions was analyzed to achieve a better network and increase detection performances.

In addition, by replacing the backbone and adopting a new network structure, the model can be accelerated. The commonly used network backbone today is MobileNet. MobileNet (MobileNetV1) [36] is a lightweight CNN network suitable for deployment in edge devices. The depthwise convolution (DSC) is used to change the convolution calculation method by reducing the number of network parameters, which balances the detection accuracy and speed. Subsequently, MobileNetV2 [37] adds two new features: The inverted residuals method enables better feature transmission ability and deeper network layers; the linear bottleneck module replaces the nonlinear module and reduces the loss of low-level features. MobileNetV3 [38], released in 2019, combines some of the structures in v1 and v2, integrating, optimizing, and removing the computationally expensive network layers in the v2 architecture, and it introduces a new lightweight attention structure for SE-NET (squeeze-and-Excitation Networks) to realize low resource consumption with almost no accuracy loss. Recently, there have also been many defect detection studies based on the MobileNet model. Xia et al. [39] adopted the lightweight backbone MobileNet-V3 to classify suspicious regions of PCB and achieved state-of-the-art speed and accuracy. Similarly, Shao et al. [40] proposed a MobileNet-YOLOv3 network for PCB defect detection. Xie et al. [41] proposed a lightweight metal surface defect detection algorithm EMV2YOLOv4 based on YOLOv4, which could meet the requirements of lightweight deployment and accuracy requirements of metal surface defect detection. Similarly, Guo et al. [42] used MobileNetV3 to replace the basic network of YOLOv5 for road damage detection, which greatly reduced the number of parameters and GFLOPs of the model and reduced the size of the model. Combined with the K-Means clustering algorithm, the accuracy and model size are guaranteed.

The works of the above researchers are of great significance for magnetic ring defect detection, but there are still many problems, which are stated as follows:Most researchers use traditional algorithms to detect magnetic ring defects, which have problems such as long project cycles, complex deployment, poor algorithm robustness, and insufficient generalization.Some researchers use deep learning to detect magnetic ring defects online. Still, the number of model parameters and the amount of calculation is too large for real-time inspection requirements.A small number of researchers streamlined the model, but the pruning strategy will reduce the accuracy and generalization of the model and fail to achieve a good trade-off between detection speed, accuracy, and the number of model parameters.The original CIOU loss function of YOLOv5 is complicated and requires a long period of time to train.

Although YOLO underwent several version iterations and its performance continuously improved, the YOLOv5 version is still the most widely used in the industry. To solve the above problems, we continue to improve and experiment on the YOLOv5 model, proposing a new model that can achieve fast and accurate magnetic ring defect detection. In this paper, an improved YOLOv5 framework called MR-YOLO is applied to the defect detection of magnetic rings, improving defect detection speeds and accuracies.

## 3. Proposed Method

### 3.1. Original YOLOv5 Network

In terms of detection accuracy and speed, the YOLOv5 model is particularly helpful for defect detection and recognition, as was already mentioned [25,26,27]. Presently, YOLOv5 is being used more and more in the industrial sphere by researchers.

YOLOv5 is a single-stage target detection network. According to the network depth and feature map width, YOLOv5 can be divided into YOLOv5s, YOLOv5m, YOLOv5l, and YOLOv5x models. YOLOv5S has the fastest processing speed, and YOLOv5x has the highest detection accuracy. The four models have the same network structure, consisting of a backbone, a neck, and a head, as shown in Figure 1. Specifically, the YOLOv5 algorithm adopts a separate CNN model to realize target detection. Firstly, the input image is scaled to the same size after data enhancement and then sent to the CNN network. Secondly, the network output obtains prediction results at three different scales, each corresponding to N channels and containing the predicted information. Then, the network prediction results can be processed by NMS operations to obtain the detected target. Finally, the NMS operation can process the network prediction results to acquire the discovered target. The network’s module structure is depicted in Figure 2, where Conv2d stands for two-dimensional convolution, BN stands for batch normalization, Silu stands for activation, Upsample stands for upsampling, and SPPF stands for a modified SPP pyramid pool module. Even if the SPPF module is faster than other modules, the quantity of calculations is still huge. To solve this issue, we considered revamping the backbone of the network.

### 3.2. YOLOv5-MV3 Network Structure

In the mass production detection of the magnetic ring, even the YOLOv5s model with small computation and complexity has the problem of insufficient real-time performance. In the YOLOv5 model, the CSP-backbone is designed for multi-classification detection problems, which achieved good performance on the COCO (class 80) and PASCAL VOC (class 20) datasets. However, magnetic ring defects have few categories and low complexity. The backbone, which has strong generalization but heavy computation, is not suitable for extracting features. Figure 3 is the part of the circular defect sample, which is divided into three categories: top cracks, inner cracks, and adhesion. Usually, the crack is mainly caused by heat bilge cold shrink during the heat treatment process, and adhesion forms from the copper billet in the process of forming the adhesive of the adjacent circular. In order to make the above defects clear, each magnetic ring has two different light sources for lighting.

To realize the trade-off between the speed and accuracy of magnetic ring detection, we improve the YOLOv5s network and propose a lightweight magnetic ring detection network model based on YOLOv5s. Figure 4 shows the schematic diagram of the improved YOLOv5s structure, where the boxes in the figure represent each module of the detection model, and the arrows connect each module to form a neural network. The MobileNetV3 module (orange part) is introduced to replace the backbone of YOLOv5 to reduce the number of parameters and computation, speeding up the detection. At the same time, we further optimized the model’s volume by replacing the complex SPPF module with a lightweight SE attention module. The backbone output will enter the neck part with PanNet as the core for bidirectional feature fusion and then enter the head part to detect targets of different sizes. Finally, post-processing operations such as NMS will obtain the detection results.

The module composition of each part of the network is shown in Figure 5. The MobilenetV3 block comes from the latest version of the MobileNet series. MobileNetV3 introduces the lightweight attention structure SE-NET. As shown on the left side of Figure 4, the inverted residual structure uses point convolution to amplify the number of channels and then carries out deep convolution at a higher level. After selecting feature channels using the SE module, point convolution is used to reduce the channels. The network uses smaller input and output dimensions, which significantly reduces the network’s computational consumption and parameter volume.

### 3.3. Extended SE Attention Module

Recent studies have shown that optimizing spatial dimension information can improve the performance of the network [43]. The SE attention module added allows the network to calibrate the importance of the channel adaptively. The attention mechanism module and the feature fusion module are added in these applications to address the issues of variable direction, sample imbalance, complex background, and difficulty in detecting small objects in visual inspection, and the modules offer useful methods for magnetic ring detection.

The SE attention module mainly contains Squeeze and Excitation [44]. Global spatial information is collected in the Squeeze module by global average pooling. The Excitation module captures channel-wise relationships and outputs an attention vector by using fully connected layers and non-linear layers (ReLU and sigmoid). The structure diagram of the SE module is illustrated in Figure 6. W and H represent the width and height of the feature map. C indicates the number of channels. Then, the input feature map size is W×H×C. The SE module first performs a Squeeze operation on the complex feature map to obtain the channel-level global features. Next, it performs an Excitation operation on the global features to learn the relationships between the various channels and to obtain their weights. Finally, it multiplies the initial feature map to obtain the final features.

### 3.4. Data Enhancement

Most original YOLOv5 data enhancement techniques, including cutoff, random erase, and the Mosaic data enhancement technique, are still used in this study. However, the rectangle reasoning data augmentation strategy has been abandoned due to the experiment’s subpar training results. Electromagnetic fields and mechanical vibrations introduce noise to the image. The cost of the project will increase if more sophisticated anti-interference hardware is used. As a result, the mosaic data enhancement method is a good alternative.

To further enhance the dataset, speed up network training, and improve the ability to recognize small targets for the original data, in the training model stage, YOLOv5 utilizes the Mosaic data augmentation method, which is based on the CutMix data augmentation method [45]. According to random scaling, random cropping, and random arrangement, CutMix stitches two images, while the Mosaic data augmentation approach stitches four images, as illustrated in Figure 7. We demonstrate the approach using a few images from the PASCAL VOC dataset due to the modest variations in magnetic ring images. Firstly, four photos are randomly chosen from a batch of the training dataset. Then, these four images are then randomly scaled, flipped, and subjected to various processes before being delivered to the neural network for processing.

Figure 8 is the Mosaic training results in a particular iteration of the training process in which most of the labels can be retained. Meanwhile, the data enhancement method enriches the training dataset, and a GPU can produce more significant benefits, reducing the requirement for a large mini-batch size.

### 3.5. Loss Function Design

The loss function of YOLOv5 is composed of the following three types of loss functions: classification loss, location loss, and confidence loss. The calculation formula is as follows.
(1)Loss=lobj+lcls+lbox

The confidence loss function can be defined as the target confidence loss $L_{obj}$, which is given by the following Equation:(2)$$\begin{array}{cc} \; & L_{obj}=-\sum_{i=1}^{K*K}\sum_{j=1}^{B}I_{ij}^{obj}\left[C_{i}^{j}\log\left(C_{i}^{\prime j}\right)+\left(1-C_{i}^{j}\right)\log\left(1-C_{i}^{\prime j}\right)\right]\\ \; & -\lambda_{\mathrm{noobj}\;}\sum_{i=1}^{K*K}\sum_{j=1}^{B}I_{ij}^{noobj}\left[C_{i}^{j}\log\left(C_{i}^{\prime j}\right)+\left(1-C_{i}^{j}\right)\log\left(1-C_{i}^{\prime j}\right)\right] \end{array}$$
(3)$$\begin{array}{cc} \; & L_{cls}=-\sum_{I=1}^{K*K}I_{ij}{}^{obj}\sum_{c\in \;\mathrm{classes}\;}\left\{P_{i}{}^{j}\left(c\right)\right.\\ \; & \left.\left[1-P_{i}{}^{j}\left(c\right)\right]\log\left[1-P_{i}^{\prime }\left(c\right)\right]\right\} \end{array}$$
where $K^{*}\;K,\;B$, and $I_{ij}{}^{obj}$ are consistent with Equation (Equation 2), c is the target category, and $P_{i}{}^{j}\left(c\right)$ and $P_{i}{}^{\prime }\left(c\right)$ are the probability that the target in the JTH prediction box in the ith grid belongs to the real value and the predicted value of a certain category, respectively.

The target localization loss, $L_{loc}$, replaces the original CIoU loss function with SIoU, which redefines the penalty index considering the vector angle between the required regressions. That is, the total number of degrees of freedom is effectively reduced by adding an angle penalty cost. Applied to traditional neural networks and datasets, it shows that SIoU improves the speed of training and the accuracy of inference. The calculation of the SIoU loss function is divided into three parts: angle calculation, distance calculation, and shape calculation:(4)$$\begin{array}{c} \mathrm{IoU}=\frac{\left|B\cap B^{GT}\right|}{\left|B\cup B^{GT}\right|} \end{array}$$
where Equation (Equation 4) represents the calculation of IOU, B is the area of the predicted annotated box, $B^{GT}$ is the area of the marked real box, and the ratio of the intersection of the predicted annotated box and the real annotated box and the union of the predicted annotated box and the real annotated box is the value of IOU.
(5)$$\begin{array}{c} \Delta =\sum_{t=x,y}1-e^{-\gamma \rho_{t}} \end{array}$$

In Equation (Equation 5), Δcontains the angle calculation and distance calculation of SIOU, and it is the final result of angle calculation. x,y is the angle sine value of the center points of the real box and the predicted box in the figure, where ρ is the square of the ratio of the relative distance between the center points of the real box and the predicted box on the *X*-axis and the *Y*-axis relative to the width and height of their minimum external rectangle, and e is the Euler constant.
(6)$$\begin{array}{c} \Omega =\sum_{t=w,h}{\left(1-e^{-\omega t}\right)}^{\theta } \end{array}$$

Equation (Equation 6) is the calculation of the Ω shape, where w and h are the width and height of the prediction frame, respectively; θ is the attention coefficient in the shape calculation formula. Finally, SIOU-LOSS consists of the above three parts, and the formula is stated as follows.
(7)$$\begin{array}{c} L_{box}=1-IoU+\frac{\Omega +\Delta }{2} \end{array}$$

## 4. Experimental Setup and Method Validation

The main training steps in the magnetic ring surface defect detection experiment are as follows. Firstly, the labeled lithium battery dataset is inputted into the network model. Secondly, the initialization parameters of the model training are selected to start the network’s model training. After the training, a weight file is generated to save the model information. Finally, the weight file is loaded into the network model for image detection.

### 4.1. Data Preparation

The experimental dataset is constructed from the magnetic ring photos taken by the magnetic ring production factory, and the pixel size of the collected images is 2048 × 1536. The defective magnetic ring pictures were manually screened out, and the initial data volume was 445. In order to improve the training effect of the model, the data set is expanded to 1225 pictures using symmetrical operations, and the collected pictures are manually marked with LabelImg software. The defects to be detected mainly include top cracks, inner cracks, and adhesion. The expanded data set is randomly divided into three parts: training data set, validating data set, and test dataset according to the ratio of 8:1:1. The distribution of defects on the surface of the magnetic ring is shown in the Figure 3. Most defects are small in size, which puts forward higher requirements for the model to detect small defects.

### 4.2. Experimental Platform

The experimental platform is shown in Table 1, which consists of a workstation driven by Windows 10.

### 4.3. Network Parameter Settings

In the training process, the default hyperparameter settings are shown in Table 2.

### 4.4. Evaluation Index

The standard evaluation indices are chosen for quantitative evaluations in order to assess the model’s performance, such as P (precision), R (recall), mAP@0.5, and mAP@0.5:0.95. Moreover, we also choose the other parameters as the evaluation basis, such as average detection and processing time, the amount of parameters, FLOPs, model size, and so on. P (precision) was calculated as the ratio of the number of correctly predicted positive samples to the number of predicted positive examples, which is defined by the following.
(8)$$Precision=\frac{TP}{TP+FP}$$

The recall is the proportion of all targets that are correctly predicted, and it is governed by the following:(9)$$Recall=\frac{TP}{TP+FN}$$
where true positives (TPs) indicate the number of samples predicted by the algorithm as positive sample targets. False positives (FPs) indicate the number of samples that predict negative samples to positive ones. False negatives (FNs) represent the number of samples that the algorithm predicts as positive samples relative to negative samples. P stands for accuracy and R for recall rate.

To validate the performance of the model, the mAP is adopted as the main evaluation metric. The mAP (average precision, AP) denotes the mean average precision, which is calculated by the area of the P-R curve. The mAP takes out the AP of each category separately and then calculates the average AP of all categories. Generally speaking, the better the classifier, the higher the AP value.

In addition to detecting accuracy, another key evaluation criterion for defect detection is speed, which plays an important role in real-time scenarios. The speed is generally measured by FPS. The FPS represents the number of frames per second in which an image is detected. A higher FPS value indicates a faster detection of the system.

## 5. Experimental Results and Analysis

To validate the effectiveness of our model, we introduce a baseline YOLOv5s with a re-parameterization technique. Four ablation experiments were designed in this section, which include YOLOv5s, YOLOv5S-MobilenetV3, YOLOv5s-MobilenetV3-SE, and YOLOv5s-Mobilenetv3SE-SIOU. The experimental results are shown in Figure 9 and Table 3, respectively. We first test five different models on the same dataset, and the specific results are shown in Table 1. When the module of the backbone is replaced with MobilenetV3, compared with YOLOv5s, FLOP reduced by 60%, the number of model parameters reduced by 49.5%, the mAP reduced by 1.7%, and the model size and reasoning time reduced by 46.5% and 17.8%, respectively.

In order to improve the mAP of the model, an SE module was added to the improved YOLOv5s-MV3, and SIOU-Loss was used as the loss function. The mAP (0.5) of the final model YOLOv5s-MV3+SE+SIOU+Mocica was 1.4% higher than YOLOv5s-MV3, the average detection, and processing time; the number of FLOP and parameters slightly increased, and 2.99% reduced the model’s size. Compared with the original YOLOv5s, YOLOv5S-MV3+SE+SIOU+Mocica (MR-YOLO) can achieve 59.3% and 47.9% reductions in FLOPs and Params, respectively, a 16.6% increase in reasoning speed, and a 48.1% reduction in model size, with only a 0.3% loss in mAP.

Table 4 shows the performance comparison between Mosaic data enhancement and non-Mosaic data enhancement in each improved model. After the Mosaic data enhancement was added, map values slightly increased, and YOLOv5-MV3+SIoU+SE increased the map values by 0.3% after Mosaic data enhancement technology was used. Table 5 shows the performance of different loss functions. Experiments show that the SIoU has the best performance in this data set. Table 6 shows the comparison results between the YOLOv5s-MV3+SIoU +SE +mosaic network and other classical networks. Compared with Faster-RCNN, YOLOv3, and YOLOv3-tiny, the Yolov3-tiny network has better performance in terms of accuracy, model size, and reasoning speed. Compared with the map of YOLOv3-tiny, MR-YOLO is 3.4% higher and the number of references is 49.6% lower, which reflects the advantages of having high accuracies and the low reference number of MR-YOLO. In addition, by comparing the confusion matrix between the original YOLOv5 and MR-YOLO, as shown in Figure 10, the prediction result of MR-YOLO in the top cracks data is better than that of the original YOLOv5. The MR-YOLO is as good as the original YOLOv5 in predicting inner cracks and adhesion data. By using the comparative analysis of the confusion matrix, we obtained a more accurate and reliable result: MR-YOLO minimizes computation and model size, while maintaining appropriate accuracies.

## 6. Conclusions

In order to replace human detection and improve the accuracy and speed of magnetic ring detection, we offer an improved lightweight YOLOv5 detection model that can effectively detect cracks with quick reasoning speeds. A lightweight detection network model named MR-YOLO based on YOLOv5 is proposed. The lightweight MobileNetV3 module is introduced into the YOLOv5 network, which greatly compresses model parameters and maintains the accuracy and speed of detection. Furthermore, the SE attention module is introduced into the backbone, enabling the network in selecting better features. mosaic data enhancement methods and the addition of random noise to the training data were used to improve the generalization and robustness of the model. The SIoU loss function is used to replace the CIoU loss function to improve the positioning accuracy and regression speed of the algorithm. Experiments reveal that when compared to the YOLOv5 algorithm, the model expression effect and detection effect of the MR-YOLO algorithm is superior, with a 16.6% improvement in reasoning speed, a 48.1% reduction in model size, and almost no loss in detection accuracies. In future work, we will deploy a model on embedded devices with limited resources.

## Figures and Tables

**Figure 1 sensors-22-09897-f001:**
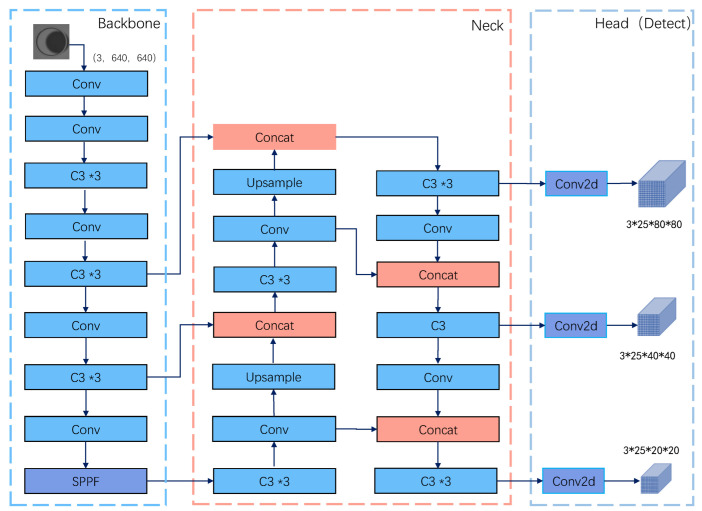
YOLOv5 network architecture. It mainly includes three parts: the backbone network, the neck, and the detection head.

**Figure 2 sensors-22-09897-f002:**
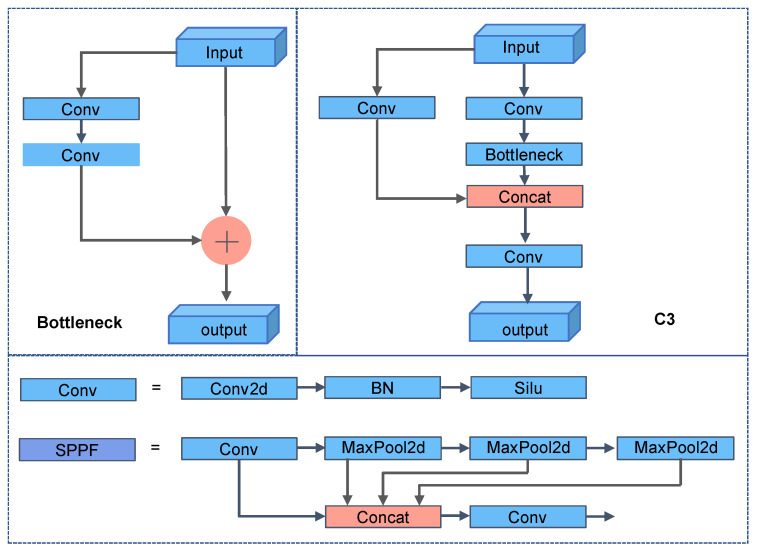
Composition diagram of each module.

**Figure 3 sensors-22-09897-f003:**
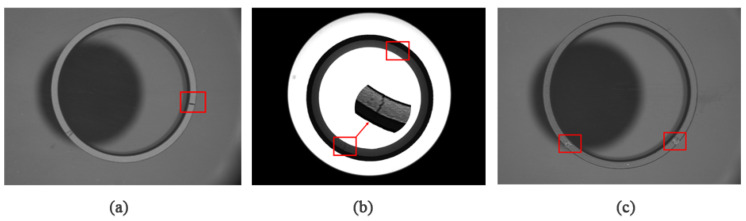
Surface defect detection results of magnetic ring based on the modified YOLOv5s algorithm: (**a**) top cracks, (**b**) inner cracks, and (**c**) adhesion.

**Figure 4 sensors-22-09897-f004:**
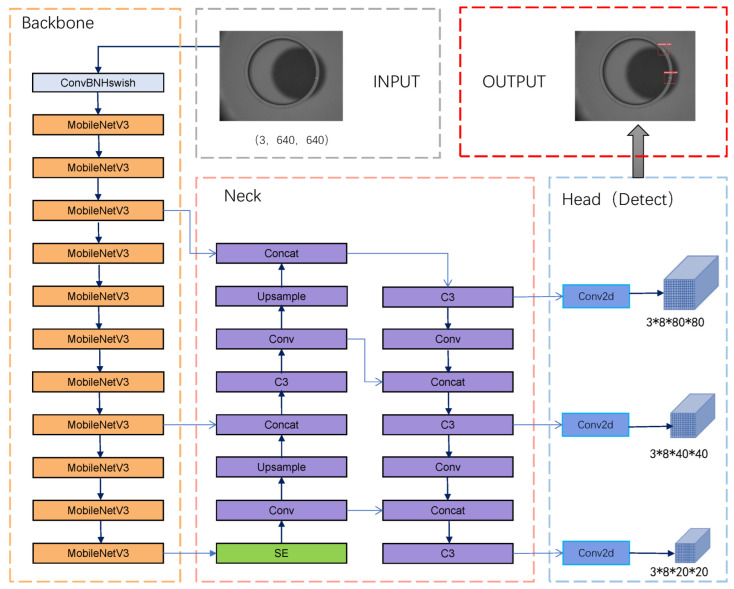
The improved YOLOv5s network architecture.

**Figure 5 sensors-22-09897-f005:**
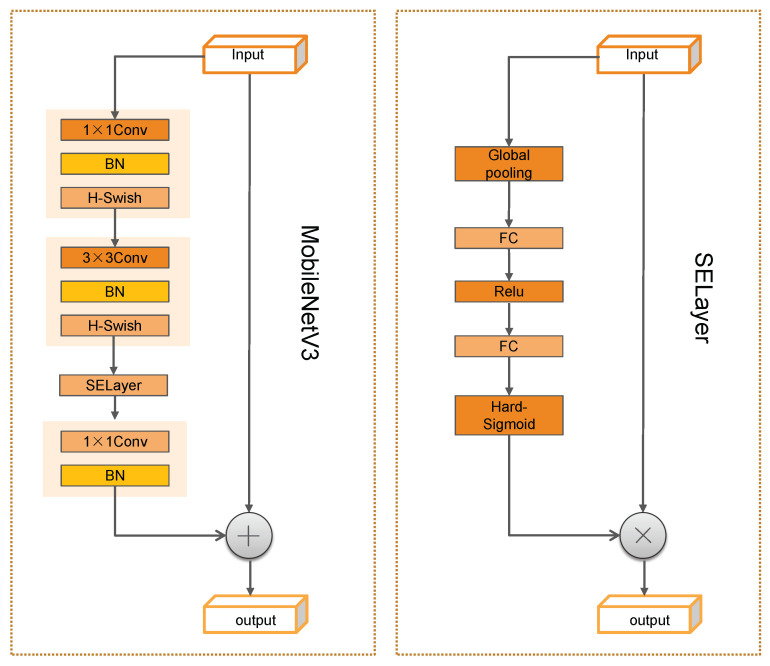
Basic composition of MobilenetV3 and the SE module.

**Figure 6 sensors-22-09897-f006:**
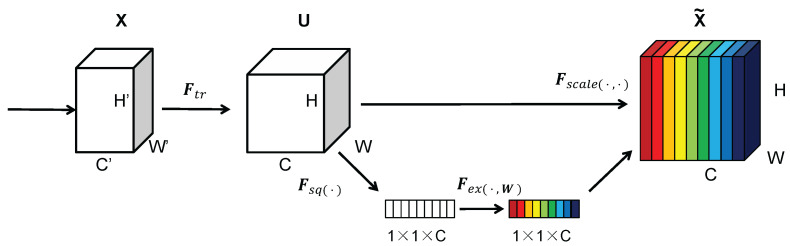
A Squeeze and Excitation block [44].

**Figure 7 sensors-22-09897-f007:**
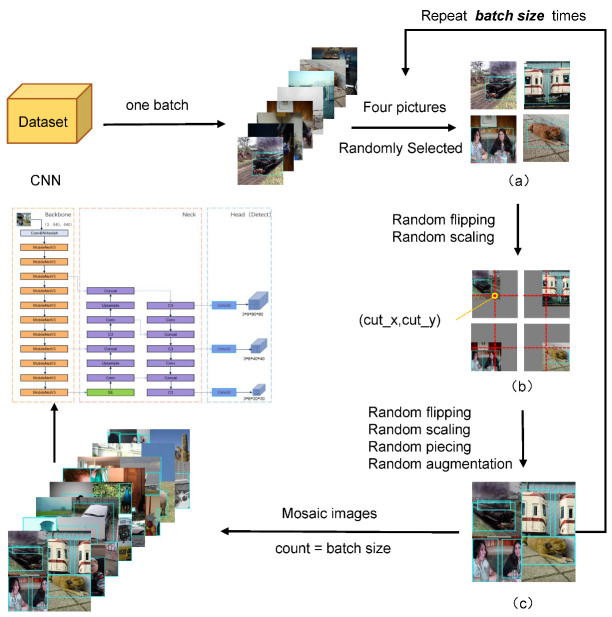
Schematic diagram of Mosaic data enhancements. (**a**) Random selecting four pictures; (**b**) Random flipping and scaling; (**c**) Random piecing and augmentation.

**Figure 8 sensors-22-09897-f008:**
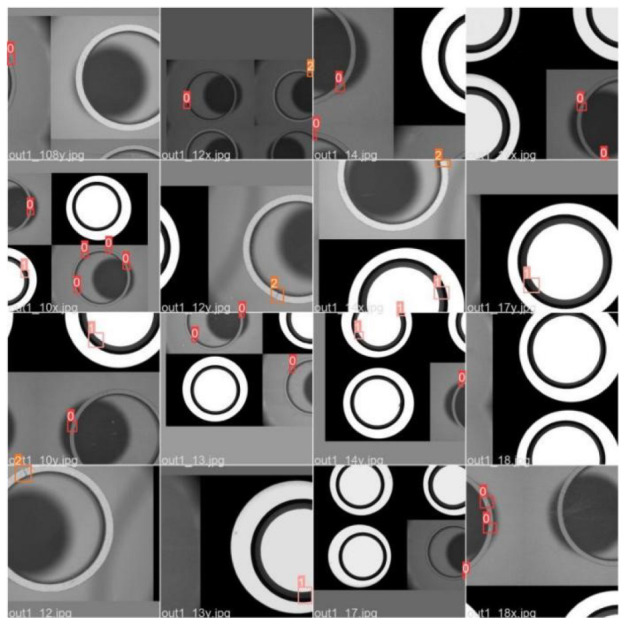
The training examples of Mosaic data enhancements.

**Figure 9 sensors-22-09897-f009:**
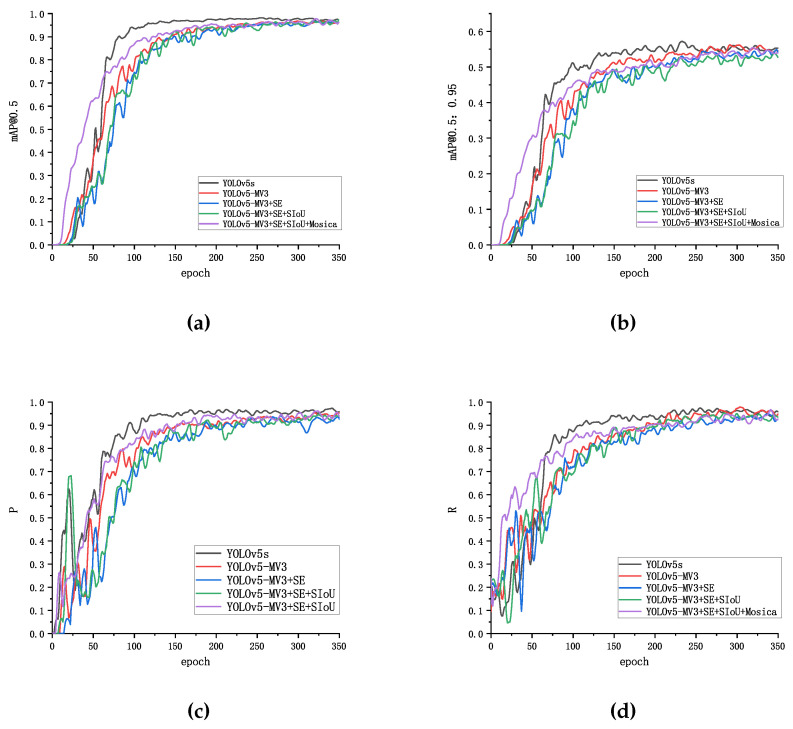
Results of different models for training. (**a**) mAP (0.5) curves of different model; (**b**) mAP (0.5:0.95) curves of different model; (**c**) precision curves of different model; (**d**) recall curves of different model.

**Figure 10 sensors-22-09897-f010:**
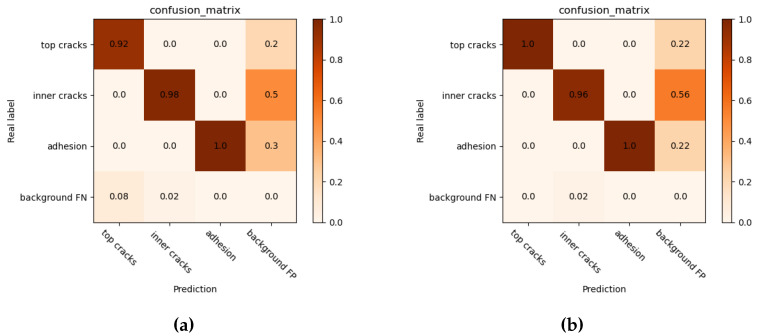
The comparison of the normalized confusion matrix between YOLOv5 and MR-YOLO. (**a**) The confusion matrix of the original YOLOV5 model; (**b**) The confusion matrix of our MR-YOLO model.

**Table 1 sensors-22-09897-t001:** Software and hardware platforms and their corresponding versions.

Hardware and Software Platform	Name
CPU	Intel I9-12900K 24
GPU	NVIDIA GeForce RTX 3090
CUDA	11.0
CUDNN	8.0
Python	3.8
Operating System	Windows 10
Deep learning Framework	Pytorch 1.8

**Table 2 sensors-22-09897-t002:** List of training parameters.

Network Parameter	Value
Training Step	800
Batch Size	64
Input Size	640 × 640
Initial Learning Rate	0.01
Momentum Decay	0.937
Weight Decay	0.0005
Training Threshold Of Confifidence	0.2

**Table 3 sensors-22-09897-t003:** Results of ablation experiments.

Model	mAP (0.5:0.95)	mAP (0.5)	FLOPs (G)	Speed-GPU (ms)	$\mathrm{Params}(10^{6})$	Weight (M)
YOLOv5s	58.1%	98.3%	16.0	24.1	7.0	13.75
YOLOv5s-MV3	54.4%	96.6%	6.4	19.8	3.6	7.35
YOLOv5s-MV3+SE	56.4%	97.3%	6.5	19.9	3.6	7.13
YOLOv5-MV3+SIoU+SE	55.2%	97.7%	6.5	19.9	3.6	7.13

**Table 4 sensors-22-09897-t004:** The result of ablation experiments with the Mosaic data enhancement.

Model	mAP (0.5:0.95)	mAP (0.5)	FLOPs (G)	Speed-GPU (ms)	$\mathrm{Params}(10^{6})$	Weight (M)
YOLOv5s	58.1%	98.3%	16.0	24.1	7.0	13.75
YOLOv5s+ Mosaic	58.3%	98.4%	16.0	24.1	7.0	13.75
YOLOv5s-MV3	54.4%	96.6%	6.4	19.8	3.6	7.35
YOLOv5s-MV3+ Mosaic	54.7%	96.8%	6.4	19.8	3.6	7.35
YOLOv5-MV3+SIoU+SE	55.2%	97.7%	6.5	19.9	3.6	7.13
YOLOv5-MV3+SIoU+SE+ Mosaic	57.1%	98.0%	6.5	19.9	3.6	7.13

**Table 5 sensors-22-09897-t005:** Training results of different loss functions of YOLOv5-MV3.

Model	mAP (0.5:0.95)	mAP (0.5)	FLOPs (G)	Speed-GPU (ms)	$\mathrm{Params}(10^{6})$	Weight (M)
YOLOv5-MV3+CIoU	54.4%	96.6%	6.4	3.5	19.8	7.35
YOLOv5-MV3+EIoU	57.2%	96.2%	6.5	3.6	19.9	7.13
YOLOv5-MV3+SIoU	56.0%	97.7%	6.5	3.6	20.1	7.13

**Table 6 sensors-22-09897-t006:** Comparison with other network training results.

Model	mAP (0.5:0.95)	mAP (0.5)	FLOPs (G)	Speed-GPU (ms)	$\mathrm{Params}(10^{6})$	Weight (M)
Faster-rcnn	46.0%	96.1%	78.9	8.6	98.7	629.3
YOLOv3	57.5%	97.4%	154.6	6.1	21.0	470
YOLOv3-tiny	55.2%	94.4%	12.9	8.6	20.1	16.6
YOLOv5s-MV3+SIOU +SE +Mosaic	57.1%	98.0%	6.5	3.6	19.9	7.13

## Data Availability

Not applicable.

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
