# Peer review of "MR-YOLO: An Improved YOLOv5 Network for Detecting Magnetic Ring Surface Defects"

_sensors, 2022, doi:10.3390/s22249897_

Round 1

Reviewer 1 Report

The work is related to detect defects in magnetic rings during the production  process. As the manual detection is not that much affective being costly and inaccurate, the authors have proposed MY-YOLO based model to detect the defects in a better way by replacing MobilenetV3 and SE attention mechanism SIOU-loss function. The research work also adopts Mosica Data Techniques to enhance the model’s performance. Some of the detailed comments and recommended changes are mentioned below as the paper cannot be accepted in its current form, the changes are highly recommended and would be appreciated.

Some of the Comments are mentioned below:

·         The diagrams are well made, however it is recommended to apply a bit light background color(color of some blocks in dark colors) in blocks having text to make it more clear and better readable.

·         The writing of the paper can be improved further to enhance the quality of the draft.

·         The contributions mentioned at line 78, 79 can be written better. Recommendation > (by replacing the backbone instead of by replace the backbone). Please go through the draft to correct any such mistakes.

·         Formatting needs improvement e.g. word spacing in few lines having references need to be adjusted(e.g. Line 92, 93 and 95 and many others).

·         In Section “1.1. Traditional defect detection methods”, it is recommended to highlight the weak points of the traditional methods and the gaps that occur.

·         Equation 4 needs a bit elaboration about it and details/ discussion of parameters used in it.

·         Network Parameter Settings and any such configurations or settings are encouraged to be mentioned in Tabular Form(other than in text).

·         Discussion on Results is very limited, recommendation is to elaborate more.

·         Improvement in Conclusion Section is recommended.

Author Response

Dear Editor,

Manuscript ID: sensors-2088080

We are very grateful for your and the reviewers’ critical comments and thoughtful suggestions. Based on these comments and suggestions, we have made modifications to the original manuscript. All changes made to the text are marked in the “marked” revision copy so that they can be easily identified and the “clean” revision copy (without marked changes) is also provided. Your questions were answered below.

Once again, we acknowledge your and the reviewers’ comments and constructive suggestions very much, which are valuable in improving the quality of our manuscript.

Kind regards,

Mr. Shuangbao Shu

Here are our responses to your comments point-by-point.

  1. The diagrams are well made, however, it is recommended to apply a bit of light background color(the color of some blocks in dark colors) in blocks having text to make it more clear and better readable.

Response:  We have re-drawn these blocks in Figures 1, 2, 4, and 5 to make it more clear and better readable.

  1. The contributions mentioned in lines 78, and 79 can be written better. Recommendation > (by replacing the backbone instead of by replacing the backbone). Please go through the draft to correct any such mistakes.

Response:  We have corrected similar errors and marked them in the revised paper.

  1. The formatting needs improvement e.g. word spacing in a few lines having references needs to be adjusted(e.g. Line 92, 93, and 95 and many others).

Response:  We have modified this problem and similar mistakes. Thank you very much.

  1. In Section “1.1. Traditional defect detection methods”, it is recommended to highlight the weak points of the traditional methods and the gaps that occur.

Response: We have added some new comments to this paragraph that focus on the inadequacies and gaps in the traditional algorithm.

  1. Equation 4 needs a bit of elaboration about it and details/ discussion of the parameters used in it.

Response: We have decomposed the original Equation 4 into four formulas and explained it in detail.

  1. Network Parameter Settings and any such configurations or settings are encouraged to be mentioned in Tabular Form(other than in text).

Response: We have represented the contents of 3.2 Experimental Platform and 3.3 Network Parameter Settings in table format.

  1. Discussion on Results is very limited, the recommendation is to elaborate more.

Response: In the result part of this paper, the confusion matrix comparison between MR-YOLO and the original YOLOV5 is added. By comparing the confusion matrix between the original YOLOV5 and MR-YOLO, the prediction result of MR-YOLO in the top cracks data is better than that of the original YOLOV5.  MR-YOLO is as good as the original YOLOV5 in predicting inner cracks and adhesion data. Further, MR-YOLO still has the same excellent accuracy as the original YOLOV5, while ensuring a small amount of computation and model volume. Through the above comparative analysis, the results are enriched and persuasive.

  1. Improvement in Conclusion Section is recommended.

Response: We have rewritten the conclusion. It focuses on a series of improvements we have made to MR-YOLO, as well as performance improvements. It mainly includes: The lightweight mobilenetv3 module is introduced into the yolov5 network, which greatly compresses model parameters and maintains the accuracy and speed of detection. Meanwhile, the SE attention module is introduced into the Backbone, enabling the network to better select features. Mosica data enhancement method and the addition of random noise to the training data were used to improve the generalization and robustness of the model. The SIoU loss function is used to replace the CIoU loss function to improve the positioning accuracy and regression speed of the algorithm. The future work content is proposed to deploy the MR-YOLO model to the platform with limited computing power.

That’s all. Thank you very much.

Reviewer 2 Report

1. How the magnetic interferences eliminated in these networks

2. How YOLOv5 networks is  useful to detect and elimination of magnetic effect.

3. Rewrite abstract and conclusion and rearrange the article in the better way

4. Add some latest article in the review

5. Enhance the result part and present it properly

Author Response

Dear Editor,

Manuscript ID: sensors-2088080

We are very grateful for your and the reviewers’ critical comments and thoughtful suggestions. Based on these comments and suggestions, we have made modifications to the original manuscript. All changes made to the text are marked in the “marked” revision copy so that they can be easily identified and the “clean” revision copy (without marked changes) is also provided. Your questions were answered below.

Once again, we acknowledge your and the reviewers’ comments and constructive suggestions very much, which are valuable in improving the quality of our manuscript.

Kind regards,

Mr. Shuangbao Shu

Here are our responses to your comments point-by-point.

  1. How the magnetic interferences eliminated in these networks

Response:   In this paper, the scratches, pits, stains, and other defects on the surface of the magnetic ring produced in the production process are detected by the method of visual inspection(Simply put, it is to obtain the picture of the magnetic ring surface through the industrial camera, according to the collected photos for inspection), which will not be affected by the magnetic effect. Or whether we misunderstood your meaning?

  1. How YOLOv5 networks are useful to detect and eliminate magnetic effects.

Response: As you see from our answer to the first question, there is no interference during the detection process. And our model can eliminate the noise of the images to find the cracks correctly by using the MR-YOLO. That’s all. Thank you.

  1. Rewrite the abstract and conclusion and rearrange the article in the better way

Response: We have rewritten the abstract and conclusion, and rearranged the paper in a reasonable format. And thank you.

  1. Add some latest articles in the review

Response: I have added three relevant articles published in this year, which are inserted on line 85, line 88, and line 186 in the revised article.

  1. Enhance the result part and present it properly

Response: In the result part of this paper, the confusion matrix comparison between MR-YOLO and the original YOLOV5 is added. By comparing the confusion matrix between the original YOLOV5 and MR-YOLO, the prediction result of MR-YOLO in the top cracks data is better than that of the original YOLOV5.  MR-YOLO is as good as the original YOLOV5 in predicting inner cracks and adhesion data. Furthermore, MR-YOLO still has the same excellent accuracy as the original YOLOV5, while ensuring a small amount of computation and model volume. Through the above comparative analysis, the results are enriched and persuasive.

That’s all. Thank you very much.

Round 2

Reviewer 1 Report

In a much better form now. Accepted from my side.